# Iron Deficiency in Cystic Fibrosis: A Cross-Sectional Single-Centre Study in a Referral Adult Centre

**DOI:** 10.3390/nu14030673

**Published:** 2022-02-05

**Authors:** Hervé Lobbes, Stéphane Durupt, Sabine Mainbourg, Bruno Pereira, Raphaele Nove-Josserand, Isabelle Durieu, Quitterie Reynaud

**Affiliations:** 1Service de Médecine Interne, Hôpital Estaing, CHU de Clermont-Ferrand, F-63000 Clermont-Ferrand, France; 2SIGMA Clermont, Institut Pascal, CHU Clermont-Ferrand, Université Clermont Auvergne, CNRS, F-63000 Clermont-Ferrand, France; 3Département de Médecine Interne et Centre de Référence Mucoviscidose, Centre Hospitalier Lyon Sud, Hospices Civils de Lyon, F-69310 Pierre-Bénite, France; durupt@chu-lyon.fr (S.D.); sabine.mainbourg@chu-lyon.fr (S.M.); raphaele.nove-josserand@chu-lyon.fr (R.N.-J.); isabelle.durieu@chu-lyon.fr (I.D.); quitterie.reynaud@chu-lyon.fr (Q.R.); 4Equipe Evaluation et Modélisation des Effets Thérapeutiques, UMR 5558, Laboratoire de Biométrie et Biologie Evolutive, CNRS, Claude Bernard University Lyon 1, F-69622 Villeurbanne, France; 5Biostatistics Unit, Centre Hospitalier Universitaire de Clermont-Ferrand, F-63000 Clermont-Ferrand, France; bpereira@chu-clermontferrand.fr; 6Research on Healthcare Performance (REHSAPE), INSERM U1290, Université Claude Bernard Lyon 1, F-69373 Lyon, France

**Keywords:** iron deficiency, cystic fibrosis, anaemia, ferritin

## Abstract

Iron deficiency (ID) diagnosis in cystic fibrosis (CF) is challenging because of frequent systemic inflammation. We aimed to determine the prevalence and risk factors of ID in adult patients with CF. We conducted a single-centre prospective study in a referral centre. ID was defined by transferrin saturation ≤16% or ferritin ≤20 (women) or 30 (men) μg/L, or ≤100 μg/L in the case of systemic inflammation. Apparent exacerbation was an exclusion criterion. We included 165 patients (78 women), mean age—31.1 ± 8.9 years. ID prevalence was 44.2%. ID was significantly associated with female gender (58.9% vs. 38%), lower age (29.4 ± 8.5 vs. 32.5 ± 9.1), lower body mass index (20.5 ± 2.2 vs. 21.3 ± 2.5), and *Pseudomonas aeruginosa* colonization (70.8% vs. 55.1%). Diabetes mellitus, antiacid drug use and low pulmonary function were more frequent in patients with ID with no statistical significance. The use of CFTR correctors was not associated with ID. In the multivariate analysis, ID was associated with female gender (OR 2.64, CI95% 1.31–5.31), age < 30 years (OR 2.30, CI95% 1.16–4.56), and *P. aeruginosa* (OR 2.09, CI95% 1.04–4.19).

## 1. Introduction

Iron is an essential micronutrient ensuring several vital body functions [1]. Iron deficiency (ID) has been frequently reported in cystic fibrosis (CF) and is attributed to a combination of chronic inflammation, impaired dietary iron absorption, malnutrition, and increased iron loss via sputum [2]. A specific concern regarding iron homeostasis in cystic fibrosis is its association with colonization by *Pseudomonas aeruginosa*, which is a well-known compounding factor of patients’ respiratory conditions [3]. In addition, there is a fear that iron supplementation may promote *P. aeruginosa*-related infections, as iron has been shown to increase bacterial growth in vitro [4]. 

Most of the data on the prevalence of iron deficiency in CF come from previous studies, before advances were made in the treatment of CF exacerbation, and especially in the improved nutritional management of patients with CF (pwCF) [2,5,6,7,8,9]. A recent study investigated the iron store status in an adult cohort of pwCF, revealing a prevalence of 41.8%, which was associated with anaemia and poor lung function [10].

In a general setting, assessing the serum ferritin level is the most sensitive and specific test for ID diagnosis, with a threshold of 20 μg/L (women) or 30 μg/L (men), providing 92% sensitivity and 98% specificity. Other biological markers can occasionally be used, such as transferrin saturation (TSAT, threshold < 16%) [11]. The main challenge in ID diagnosis for pwCF is the effect of systemic inflammation on biological iron markers. Inflammation leads to an increase in ferritin levels due to the enhanced hepcidin production through the effects of interleukin-6 [12]. In pwCF studies, very heterogeneous biological definitions of iron deficiency coexist, probably related to the concurrent recruitment of adult and paediatric populations.

Therefore, we aimed to identify the prevalence of ID in a recent homogeneous cohort of adult pwCF, using the most common biological definitions. The secondary objective of our study was to identify the risk factors for iron deficiency.

## 2. Materials and Methods

We conducted an observational cross-sectional single-centre study. Patients were screened for participation during their annual medical systematic visit to the Referral Cystic Fibrosis Centre in Lyon, Hospices Civils, France.

### 2.1. Patients

The inclusion criteria were: (i) genetically proven CF, (ii) age ≥ 18 years, and (iii) serum iron parameter assessment. The exclusion criteria were: (i) current or past use of dietary iron supplement or iron supplement drugs (oral or intravenous) in the previous year, and (ii) clinically apparent exacerbation of CF between the previous 7 days and the day of the medical consultation, defined by one or more of the following symptoms (Fuchs criteria [13]:-New or increased cough, sputum production or chest congestion.-Decreased exercise tolerance, increased dyspnoea.-Increased fatigue, decreased appetite.-Increase respiratory rate or dyspnoea at rest.-Change in sputum appearance.-Fever.

### 2.2. Data Collection

Baseline data were extracted from medical records: gender, CFTR mutations, history of diabetes mellitus, solid organ transplantation, oxygen requirement, current treatment, including CFTR modulators, antiacids (alginate, anti-H2 and proton pump inhibitors: PPI), pancreatic enzymes, and vitamin supplement (A, D, K and E). 

The following parameters were prospectively collected the day of the annual medical check-up: (i) clinical data: age, weight, height, and body mass index (BMI); (ii) biological data: bronchial colonization, iron stores (ferritin, TSAT), C reactive protein (CRP), liver function tests, creatinine, complete blood count, blood glucose, albumin, vitamin level (A, D, E, K), and (iii) pulmonary function test (PFT), including the measurement of forced expiratory volume in one second (FEV1) and forced vital capacity (FVC).

Sputum samples were collected the day of inclusion through physiotherapy manoeuvres (oscillatory positive expiratory pressure and forced expiration with acceleration of expiratory flow) and sent to the microbiology department for culture to search for *P. aeruginosa* colonization. 

The main laboratory assays used during the study were: quantitative immunoturbidimetric (Architect, Abbott) for C reactive protein (normal < 5 mg/L) and transferrin (normal ranges: 1.74–3.64 g/L), chemiluminescent microparticle immunoassay (Architect, Abbott) for ferritin (normal ranges: 22–275 μg/L), and direct FERENE photometry (without deproteinization—Architect, Abbott) for serum iron (12.0–31.0 μmol/L).

### 2.3. Anaemia and Micronutrient Deficiency Definitions

Anaemia was defined using the WHO criteria [14]: haemoglobin < 120 (women) or 130 (men) g/L.

Two definitions of ID were studied in our protocol:(i)The international recommended biological definition of ID [11] was used as our primary endpoint: ferritin ≤20 (women) or 30 (men) μg/L, or ≤100 μg/L in the case of systemic inflammation (CRP ≥ 10 mg/L) or TSAT ≤ 16%.(ii)The historical paediatric CF definition of ID [5,15,16] was used as a secondary endpoint: ferritin ≤ 12 μg/L or TSAT ≤ 16%.

We also studied a population with mildly depleted iron stores, defined by ferritin ≤ 50 μg/L, as several studies showed an improvement of life quality parameters after intravenous iron supplementation in this population [17]. 

Fat-soluble vitamin deficiencies were defined as follows: vitamin K < 1000 ng/L, vitamin D < 30 ng/mL [18], vitamin E < 12 μmol/L [19], and vitamin A < 0.52 μmol/L [20]. 

### 2.4. Statistics

Categorical data were described with numbers (percentages). Continuous data were expressed as mean and standard deviation or median and interquartile range, according to the statistical distribution. The assumption normality of the data was assessed using the Shapiro-Wilk test. The comparisons between independent groups according to ID were performed using the Student’s *t*-test or the Mann–Whitney test when the assumptions of the *t*-test were not met. Homoscedasticity was assessed using the Fisher–Snedecor test. Categorical data were compared between groups (ID yes/no) using chi-squared or Fisher’s exact tests. 

Then, a multivariable analysis was conducted to determine the factors associated with ID. Generalized linear modelling (logistic for binary dependent outcome: ID yes/no) was performed with covariates fixed according to the univariate results and to the clinical relevance, with particular attention paid to multicollinearity. Furthermore, age and BMI, which did not follow a Gaussian distribution, were categorized according to their statistical distribution. The results were expressed using odds ratios (OR) and 95% confidence intervals. 

To ensure the robustness of our results, the final model was validated by a two-step bootstrapping process. In each step, 1000 bootstrap samples with replacements were created from the training set. In the first one, using the stepwise procedure, we determined the percentage of models, including each of the initial variables. In the second step, we independently estimated the logistic model parameters of the final model. The bootstrap estimates of each covariate coefficient and standard errors were averaged from these replicates. Statistical analyses were performed using Stata 15 (StataCorp, College Station, TX, USA). All statistical tests were two sided, with a type I error set at 5%.

### 2.5. Ethics

The study was conducted in accordance with the Declaration of Helsinki and registered on https://clinicaltrials.gov/ct2/show/NCT04584489 (accessed on 14 October 2020). According to French regulations, patient consent was waived. The study protocol was approved by the local International Review Board (69HCL20_0793).

## 3. Results

We assessed 226 PwCF from 6 October 2020 to 23 February 2021 for eligibility. Figure 1 provides a STROBE-compliant flow diagram [21].

### 3.1. Baseline Characteristics

Table 1 summarizes the characteristics of the 165 patients (78 women, 47%) included in the final analysis. The mean age was 31.1 ± 8.9 years. The types of CFTR mutations are available as Appendix A. The most frequent CF complication was diabetes mellitus (27/165, 16.3%), which was significantly more frequent among women. Regarding solid organ transplantation, 14/165 (8.5%) had lung transplants (median delay—76 months), two patients had liver transplants, and one patient had a kidney transplant. In total, 57/165 (34.5%) patients were treated by CFTR correctors: ivacaftor (*n* = 3), ivacaftor/lumacaftor (*n* = 42), ivacaftor/tezacaftor (*n* = 1), and ivacaftor/tezacaftor/elexacaftor (*n* = 11). 

The mean CRP was 6.8 ± 11.4 mg/L. Thirty-two patients had significantly increased CRP > 10 mg/L, showing systemic inflammatory conditions. The prevalence of vitamin A, D, E and K deficiency was 3/142 (2%), 19/137 (13%), 11/142 (7.7%), and 33/141 (23%), respectively.

### 3.2. Iron Deficiency Prevalence

A total of 73 of the 165 patients (44.2%) pwCF had ID, according to the definition retained for our primary endpoint. Among them, only 9/73 were classified as ID because the ferritin level was ≤100 μg/L with CRP ≥ 10 mg/L. 

Using the historical definition of ID in CF, 53/165 (32.1%) pwCF had ID. As illustrated in Figure 2, each patient with ID defined through the historical definition of ID was included in the primary endpoint definition. 

Finally, 82 patients (49.7%) of our cohort had mildly depleted iron stores (ferritin ≤ 50 μg/L), including 54/73 patients with ID, according to the primary endpoint definition. 

Nine patients were anaemic: seven women (four with ID) and two men (one with ID). The mean corpuscular volume (MCV) was significantly lower in iron-deficient patients (85.8 ± 5.2 vs. 88.5 ± 4.7 fL, *p* < 0.001). MCV was significantly lower in ID patients irrespective of gender (men: 85.2 ± 3.3 vs. 88.1 ± 3.6 fL, *p* < 0.01, women: 86.2 ± 6.2 vs. 89.1 ± 6.0 fL, *p* = 0.01).

### 3.3. Iron Deficiency Risk Factors

#### 3.3.1. Univariate Analysis

Table 2 shows the association of ID and risk factors. ID was significantly associated with female gender (58.9% vs. 38%, *p* = 0.008), lower age (29.4 ± 8.5 vs. 32.5 ± 9.1 years, *p* = 0.02), lower BMI (20.5 ± 2.2 vs. 21.3 ± 2.5, *p* = 0.05), and *P. aeruginosa* colonization (70.8% vs. 55.1%, *p* = 0.04). 

Diabetes mellitus and the use of antiacid drugs or pump proton inhibitors were more frequent in the ID group, but the difference did not reach statistical significance (*p* = 0.08 and *p* = 0.06, respectively). The prevalence of ID tended to be correlated with worsened PFT, illustrated by the decrease in FEV1 (*p* = 0.07). The proportion of patients treated with CFTR correctors was similar in the ID and no-ID groups (*p* = 0.60). Fat-soluble vitamin deficiency was not significantly associated with ID.

#### 3.3.2. Multivariate Analysis

In the multivariate analysis (Figure 3), female gender (OR: 2.64, CI95%: 1.31–5.31, *p* = 0.006), age below 30 years (OR: 2.30, CI95%: 1.16–4.56, *p* = 0.02), and *P. aeruginosa* (OR: 2.09, CI95%: 1.04–4.19, *p* = 0.04) were significantly associated with ID, whereas diabetes mellitus and BMI were not associated with ID.

Noticeably, we repeated the multivariate analysis including only the 64 patients defined as ID with low ferritin or TSAT level and excluding those with CRP ≤ 100 μg/L in the case of systemic inflammation (CRP ≥ 10 mg/L) or TSAT ≤ 16%. Similar results were found: ID was significantly associated with *P. aeruginosa* colonization (OR: 2.15, CI95%: 1.03–4.48, *p* = 0.04), female sex (OR: 3.38, CI95%: 1.65–6.93, *p* = 0.001), and age < 30 years (OR: 2.57, CI95%: 1.26–5.27, *p* = 0.009), whereas diabetes mellitus (OR: 1.84, CI95%: 0.72–4.69, *p* = 0.20) and BMI < 20 kg/m^2^ (OR: 0.88, CI95%: 0.43–1.78, *p* = 0.71) were not associated with ID.

## 4. Discussion

In this prospective study, we found a high prevalence of ID in adult pwCF, similar to the previous most recent study but a low prevalence of anaemia [10]. To circumvent the non-specific elevation of ferritin during systemic inflammation, we recruited stable patients during their annual evaluation and used the most recent guidelines for ID diagnosis, which recommend increasing the threshold of ferritin in the case of high CRP levels [11]. The use of serum iron concentration or soluble transferrin receptor (sTfR) has been suggested in such situations, but serum iron concentration is reduced in the case of systemic inflammation and the sTfR can be affected by enhanced erythropoiesis and has a lower sensitivity and specificity than ferritin [22]. Moreover, the use of sTfR is limited by the lack of standardization of sTfR measurement [23].

Very few data about the risk factors of ID in adult pwCF are available at present. Previous studies focused on children [7] or mixed populations (both adult and child patients [8] and were impaired by their sample size, missing data, or confounding factors [16]. Table 3 presents the main previously published studies during the past 40 years (PubMed literature review using the following mesh terms—“Iron”, “Iron deficiency” and “Cystic Fibrosis”), showing the great heterogeneity of biological definitions used to define ID in pwCF. 

Unlike the study of Gettle et al. [10], antiacid drug use was not associated with ID in our cohort, whereas a lower BMI, lower age, and female gender appeared as risk factors of ID in the multivariate analysis. These discrepancies might be due to a younger population and a lower mean BMI in our work or to the systematic inflammation assessment that allowed the reclassification of nine patients who were misdiagnosed as non-iron deficient. In previous studies [6], FEV1 was correlated with ID. A similar trend was present in our results, without statistical significance. In our study, patients were less frequently colonized by *P. aeruginosa* and mean FEV1 was higher than in previous cohorts, reflecting the improved medical management of CF. We expected a lower prevalence of ID in lung transplant patients, related to decreased systemic inflammation linked to the disappearance of *P. aeruginosa* colonization. However, the difference was not statistically significant, probably because of the small number of lung transplant recipients with ID (2/14).

ID aetiology in CF remained uncertain. The high prevalence in men compared to the general population (2% according to the Centre for Disease Control [32]) highlighted that ID in CF is not supported by the same blood loss mechanism. The treatment of CF exacerbation has been shown to be linked to an increase in iron stores, supporting the hypothesis that systemic inflammation through enhanced hepcidin secretion is a central cause of ID [26]. As such, the increasing use of CFTR modulators, which are known to reduce the systemic inflammation, appears as a promising option for preventing ID [33]. One-third of our patients received these treatments with no difference between ID and no-ID groups. A lack of power due to the sample size could explain these results: a prospective assessment of iron status before and after highly effective CFTR modulator use appears necessary. Indeed, triple combination therapy is expected to provide better control of pulmonary inflammation, which could thus help to prevent ID and anaemia [34]. 

The treatment of ID is controversial and no recommendations are available [35]. Iron supplementation has been suspected to enhance bacterial growth. In a small case series, the use of ferric carboxymaltose was associated with the worsening of pulmonary symptoms and FEV1 within a few days following infusion [36]. However, in a blinded trial, weekly intravenous ferrous sulphate administration in anaemic ID adult pwCF did not increase CF exacerbation [28,37]. Future studies should focus not only on the iron supplementation tolerance profile but also on the improvement of health quality and fatigue related to the correction of ID [38,39]

Some limitations of our study can be raised. First, our study is a single-centre design and took place in a referral centre: thus, our results might not be generalizable to the whole pwCF population. Furthermore, our results are only applicable to adult patients and specific studies should be conducted in the paediatric population. Second, although we included 40% of our cohort, the sample size may limit multiple-comparison and subgroup analyses. Third, the cross-sectional nature of our study does not allow us to study the effect of treatments, such as antibiotics or CFTR modulators, on the incidence of ID. To the best of our knowledge, our study is the largest published to date, representing a first-step analysis with moderate statistical power, providing a basis and hypothesis for future prospective studies to confirm these results in larger samples.

## 5. Conclusions

ID is highly prevalent in adult patients with CF. The current guidelines for pwCF recommend reviewing iron status annually [40], but a recent survey among CF clinicians showed that few centres are routinely screening for ID. To improve ID screening in pwCF, our results suggest that special attention should be directed to young women with low BMI and *P. aeruginosa* colonization. However, male patients would also benefit from ID screening as a much higher prevalence was identified compared to young men in the general population. The therapeutic revolution of CFTR modulators could dramatically improve the prevalence of ID in CF, but further prospective studies are required to determine the effect of inflammatory syndrome normalization on iron metabolism.

## Figures and Tables

**Figure 1 nutrients-14-00673-f001:**
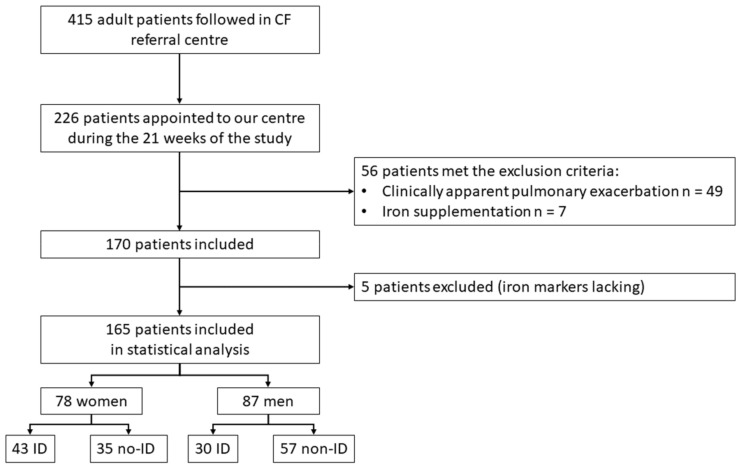
Flowchart. CF: cystic fibrosis. No-ID: no iron deficiency.

**Figure 2 nutrients-14-00673-f002:**
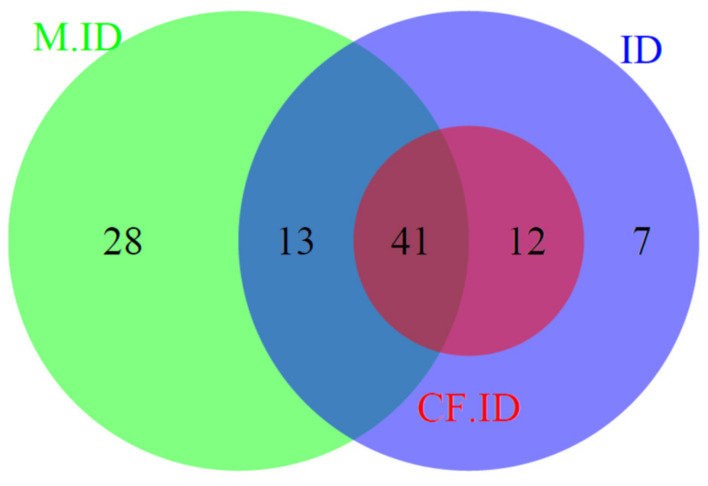
Venn diagram of iron-deficient patients, according to the different biological definitions. M.ID: patients with mild iron depletion (ferritin ≤ 50 μg/L). CF.ID: historical paediatric definition of iron deficiency in cystic fibrosis (ferritin ≤ 12 μg/L or TSAT ≤ 16%). ID: primary endpoint definition of iron deficiency, according to international criteria (ferritin ≤ 20 (women) or 30 (men) μg/L or ≤100 μg/L in the case of systemic inflammation (C reactive protein ≥ 10 mg/L) or transferrin saturation ≤ 16%).

**Figure 3 nutrients-14-00673-f003:**
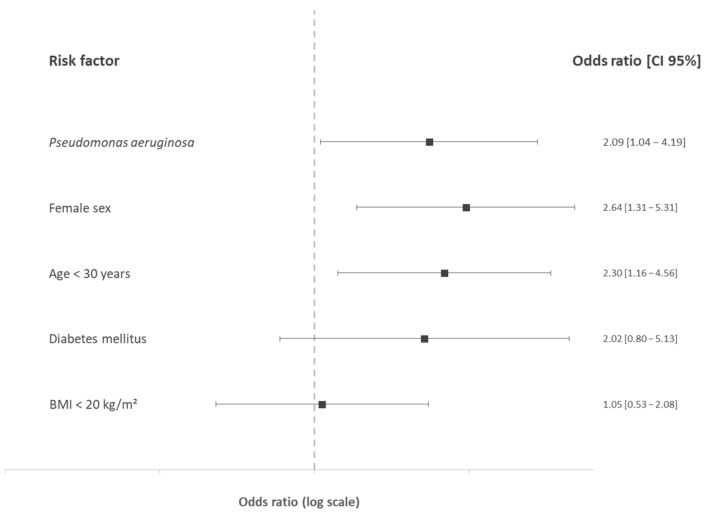
Odds ratios and 95% confidence interval of iron deficiency risk factors in cystic fibrosis in the multivariate analysis. BMI: body mass index (kg/m^2^).

**Table 1 nutrients-14-00673-t001:** Baseline characteristics of the population.

	Men (*n* = 87)	Women (*n* = 78)	*p*
Age (years)	30.2 ± 7.7	32.1 ± 10.1	0.37
Genotype			
p.PheF508del heterozygote (*n*, %)	32 (36.8%)	29 (37.2%)	0.59
p.PheF508del homozygote (*n*, %)	45 (51.7%)	36 (46.1%)	
other genotypes (*n*, %)	10 (11.5%)	13 (16.7%)	
BMI (kg·m^−2^)	21.4 ± 2.5	20.3 ± 2.1	0.005
Diabetes (*n*, %)	8 (9.2%)	19 (24.4%)	0.009
CF-related liver disease (*n*, %)	6 (6.9%)	11 (14.1%)	0.12
*P. aeruginosa* colonization (*n*, %)	54/85 (63.5%)	46/76 (60.5%)	0.69
Anaemia (*n*, %)	2/80 (2.5%)	7/67 (10.4%)	0.08
Ferritin (μg/L)	87.4 ± 69.2	49.7 ± 64.8	<0.001
TSAT (%)	22.8 ± 8.8	17.5 ± 6.1	<0.001
CRP (mg/L)	5.8 ± 8.2	8 ± 14.3	0.78
FEV1 (*n*, %)	*n* = 86	*n* = 76	
>79%	50 (58.1%)	32 (42.1%)	0.05
50–79%	17 (19.8%)	27 (35.5%)	
30–49%	19 (22.1%)	16 (21.1%)	
<30%	0	1 (1.3%)	

Data are presented as mean ± standard deviation or as number and percentage (%). CF: cystic fibrosis. CRP: C reactive protein. FEV: Forced expiratory volume in one second. n: number. TSAT: transferrin saturation.

**Table 2 nutrients-14-00673-t002:** Risk factors and 95% CI for iron deficiency in univariate analysis.

	No-ID (*n* = 92)	ID (*n* = 73)	*p*
Age (year)	32.5 ± 9.1	29.4 ± 8.5	0.02
Female gender (%)	38%	58.9%	0.008
BMI (kg/m^2^)	21.3 ± 2.5	20.5 ± 2.2	0.05
*P. aeruginosa* (%)	55.1%	70.8%	0.04
Diabetes mellitus (*n*)	11	16	0.08
CF-related liver disease (*n*)	9	8	0.80
Antiacid drugs/PPI (%)	25%	38.4%	0.06
CFTR corrector drugs (%)	32.6%	37%	0.60
FEV1			
<30%	0	1	0.07
30–49%	16	19	
50–79%	35	33	
≥80%	39	19	

BMI: body mass index; CF: cystic fibrosis; CFTR: cystic fibrosis transmembrane regulators; ID: iron-deficient; FEV1: forced expiratory volume in one second; PPI: pump proton inhibitors. NID: No iron deficiency.

**Table 3 nutrients-14-00673-t003:** PubMed literature systematic review of studies reporting iron deficiency among patients with cystic fibrosis.

	*n*	Patients	ID Biological Definition	ID Prevalence	Exacerbation
Gettle, 2020 [10]	67	A	ferritin < 12 μg/L and/or TSAT < 16%	41.8%	PEx+
Kałużna-Czyż, 2018 [24]	46	P	ferritin < 12 μg/L (<5 yo) ferritin < 15 μg/L (>5 yo)	39%	PEx+ and PEx−
Yadav, 2014 [25]	27	P	SI < 4 μmol/L	48.1%	PEx+ and PEx−
Gifford, 2012 [26]	12	A	SI < 12 μmol/L	83%	PEx+
Gifford, 2011 [27]	39	A	SI < 12 μmol/L	76.9%	PEx+ and PEx−
von Drygalski, 2008 [8]	26	A + P	SI ≤ 40 μg/dL or TSAT ≤ 20% or ferritin ≤ 35 μg/L	61% * 87.5% #	NA
Khalid, 2007 [16]	127	A	ferritin < 12 μg/L (women) and 20 μg/L (men) or SI < 12 μmol/L or TSAT ≤ 15% or sTfR < 1.74 mg/L	18.9% (ferritin) 42.5% (TSAT)15% (sTfR)	PEx+ and PEx−
Reid, 2002 [6]	30	A	SI < 12 μmol/L or TSAT < 16%	74%	PEx−
Jaffe, 2002 [28]	144	P	NA	58%	NA
Keevil, 2000 [5]	70	A	ferritin < 12 μg/L (women) ferritin < 20 μg/L (men) SI < 12 μmol/L TSAT < 16% sTfR < 1.74 mg/L	11% (ferritin) 69% (TSAT) 29% (sTfR)	NA
Pond, 1996 [29]	71	A	TSAT < 16%	62%	NA
Zempsky, 1989 [30]	13	A	ferritin ≤ 25 μg/L	38.4%	PEx−
Ehrhardt, 1987 [15]	127	A + P	ferritin < 12 μg/L	32.3%	PEx+ and PEx−
Ater, 1983 [31]	39	A + P	ferritin < 12 μg/L SI < 40 μg/dL TSAT < 16%	33% (ferritin) 25% (SI) 28% (TSAT)	PEx+ and Pex−

* among anaemic patients; # among non-anaemic patients. A: adult population; *n*: sample size; NA: not available; P: paediatric population; Pex+: included patients with clinically apparent pulmonary exacerbation; PEx−: included patients without apparent pulmonary exacerbation; SI: serum iron; sTfR: soluble transferrin receptor; TSAT: transferrin saturation.

## Data Availability

The data presented in this study are available on request from the corresponding author.

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
