# Peer review of "Iron Deficiency in Cystic Fibrosis: A Cross-Sectional Single-Centre Study in a Referral Adult Centre"

_nutrients, 2022, doi:10.3390/nu14030673_

Round 1

Reviewer 1 Report

In their article ‘Iron deficiency in cystic fibrosis: a prospective study in a modern adult cohort’ Dr. Lobbes and colleagues report the frequency and possible causes of ID in a cohort of adult CF patients.

The sample size is appropriate, the whole study is well conducted and the article well written.

I have a few points the authors should clarify:

1.

One could categorize patients with CRP < 10 mg/L and ferritin ≤ 20 µg/l for women and ≤ 30 µg/l for men as having absolute ID and patients with a CRP ≥ 10 mg/L and ferritin ≤ 100 µg/l as having a combination of absolute and functional ID.

According to this classification: Are the demographic data different between the two categories? More importantly, are the risk factors for ID (e.g. colonization with P. aeruginosa) different between the two categories?

2.

Please describe the methods and time-points for the assessment of colonization with P. aeruginosa in further detail. Have the authors detected other relevant microbes, too?

3.

Please also specify the other relevant laboratory assays used and their reference ranges e.g. for CRP, ferritin, TSAT.

4.

Please use the abbreviation P. aeruginosa throughout the manuscript, it may not be necessary to use PA as well.

5.

For FEV1, two different categories are used in the tables. Please clarify.

6.

Please check Table 1 for typos, e.g. p.PhfeF508del heterozygote and ferritine.

Other than that, the article may be of high interest for the readership of ‘Nutrients’.

Author Response

In their article ‘Iron deficiency in cystic fibrosis: a prospective study in a modern adult cohort’ Dr. Lobbes and colleagues report the frequency and possible causes of ID in a cohort of adult CF patients.

The sample size is appropriate, the whole study is well conducted and the article well written.

I have a few points the authors should clarify:

  1. One could categorize patients with CRP < 10 mg/L and ferritin ≤ 20 µg/l for women and ≤ 30 µg/l for men as having absolute ID and patients with a CRP ≥ 10 mg/L and ferritin ≤ 100 µg/l as having a combination of absolute and functional ID.

According to this classification: Are the demographic data different between the two categories? More importantly, are the risk factors for ID (e.g. colonization with P. aeruginosa) different between the two categories?

We thank the reviewer for the interesting comment. As suggested, we compared patients with absolute ID (ferritin < 20 µg/L for women and 30 µg/L for men or TSAT < 16% irrespective of gender) and functional ID (ferritin < 100 µg/L with inflammatory syndrome CRP > 10 mg/L).

64 patients had absolute ID: in univariate analysis, we found a significant association of ID with younger age (29.3 ± 8.9 vs 32.2 ± 8.8, p = 0.04), age < 30 years (p = 0.03), and female sex (p = 0.002). A trend with P.aeruginosa colonization was found (p = 0.05) and no association was found with BMI (p = 0.15) and diabetes mellitus (p = 0.13).

C reactive protein dosage was not available for 18 patients. Among the 147 patients with available CRP dosage, 15 women and 17 men had CRP > 10 mg.L-1 (mean 21.7 ± 17.5). However, among them, 12 women and 4 men were iron deficient according to their ferritin or TSAT level irrespective of the CRP level. Only 9 patients (mean CRP 22.1 ± 19.4) were diagnosed as ID according with the increase of ferritin threshold to 100 µg/L (mean ferritin 66.4 ± 24.3 µg/L).

After excluding these 9 patients, we repeated the analysis with the 64 patients with “absolute ID” (defined by ferritin < 20 µg/L for women and 30 µg/L for men or TSAT < 16% irrespective of gender) and found similar results:

  • A significant association of ID with younger age (29.3 ± 8.9 vs 2 ± 8.8, p = 0.04), age < 30 years (p = 0.03), and female sex (p = 0.002). A trend with P.aeruginosa colonization was found (p = 0.05)
  • The BMI was lower in patients with absolute ID (20.6 ± 2.3 vs2 ± 2.5) but the difference did not reach statistical significance, probably due to the decreased power with a lower sample.

Also, in multivariate analysis we identified the same risk factors for absolute ID. However, confidence intervals were larger due to the decreased statistical power with a decreased sample size:

  • aeruginosa (OR 2.15, CI 1.03-4.48, p = 0.04), female sex (OR 3.38, CI 1.65-6.93, p = 0.001) and age below 30 years (OR 2.57, CI 1.26-5.27, p = 0.009) were associated with ID.
  • Diabetes mellitus (OR 1.84, CI 0.72-4.69, p = 0.20) and BMI < 20 kg/m² (OR 0.88, CI 0.43-1.78, p = 0.71) were not associated to ID.

Moreover, the demographic data between the 64 patients with absolute ID and the 9 patients with functional ID were similar (male: female sex ratio p = 0.14, P.aeruginosa colonization p = 0.71, diabetes mellitus prevalence p = 1, mean BMI 20.59 ± 2.27 vs 20.01 ± 2.13, p = 0.59 and mean age 29.84 ± 5.17 vs 29.31 ± 8.85 p = 0.45).

To improve the readability of our manuscript we added the following paragraphs including these information’s:

Section 3.2 Iron deficiency prevalence:

“Among them, only 9/73 were classified as ID because the ferritin level was ≤ 100 µg/L with CRP ≥ 10 mg/L”

Section 3.3.2 Multivariate analysis.

“Noticeably, we repeated the multivariate analysis including only the 64 patients defined as ID with low ferritin or TSAT level and excluding those with CRP ≤ 100 µg/L in the case of systemic inflammation (CRP ≥ 10 mg/L) or TSAT ≤ 16%. Similar results were found: ID was significantly associated with P.aeruginosa colonization (OR 2.15, CI 1.03-4.48, p = 0.04), female sex (OR 3.38, CI 1.65-6.93, p = 0.001) and age < 30 years (OR 2.57, CI 1.26-5.27, p = 0.009) whereas diabetes mellitus (OR 1.84, CI 0.72-4.69, p = 0.20) and BMI < 20 kg/m² (OR 0.88, CI 0.43-1.78, p = 0.71) were not associated to ID.”

  1. Please describe the methods and time-points for the assessment of colonization with P. aeruginosain further detail. Have the authors detected other relevant microbes, too?

All the biological parameters studied were collected on the day of the patient's annual visit to the reference centre. As such, on the day of inclusion, all patients had a bacteriological examination of sputum to determine if they were colonized with P.aeruginosa.

We added the following sentence in the method section to give further details:

Sputum samples were collected the day of inclusion through physiotherapy manoeuvres (oscillatory positive expiratory pressure and forced expiration with acceleration of expiratory flow) and sent to the microbiology department for culture to search for P.aeruginosa colonization.

We detected other microbes: 96/165 patients were also colonized with S.aureus. Patients with iron deficiency were more frequently colonized with S.aureus (65% vs 55%) but the difference did not reach statistical significance (p=0.19). 54/165 patients were also colonized with other species: the most frequent were Achromobacter xylosoxidans n=13, Stenotrophomonas maltophilia n=9, non tuberculous mycobacteria n=7, Burkholderia cepacia n=7. However due to the relatively low number of patients colonized with each of these microbes no statistical association could be assessed about iron deficiency.

In order to improve the readability of the manuscript, we have chosen not to include these results in the text but we remain at the disposal of the reviewer to add them in case of contrary opinion.

  1. Please also specify the other relevant laboratory assays used and their reference ranges e.g. for CRP, ferritin, TSAT.

To provide these information’s, we added a specific paragraph at the end of the section 2.2 Data collection:

The main laboratory assays used during the study were: quantitative immunoturbidimetric (Architect, Abbott) for c reactive protein (normal < 5 mg.L-1) and transferrin (nor-mal ranges: 1.74-3.64 g.L-1), chemiluminescent microparticle immunoassay (Architect, Abbott) for ferritin (normal ranges: 22-275 µg.L-1) and direct FERENE photometry (without deproteinization – Architect, Abbott) for serum iron (12.0-31.0 µmol.L-1).

  1. Please use the abbreviation P. aeruginosathroughout the manuscript, it may not be necessary to use PA as well.

We made the correction: PA was replaced by P.aeruginosa in the manuscript.  

  1. For FEV1, two different categories are used in the tables. Please clarify.

Men (n=87)

Women (n=78)

p

Age (years)

30.2 ± 7.7

32.1 ± 10.1

0.37

Genotype

 p.PheF508del heterozygote (n,%)
 p.PheF508del homozygote (n,%)
 other genotypes (n,%)

32 (36.8%)

45 (51.7%)

10 (11.5%)

29 (37.2%)

36 (46.1%)

13 (16.7%)

0.59

BMI (kg.m-2)

21.4 ± 2.5

20.3 ± 2.1

0.005

Diabetes (n,%)

8 (9.2%)

19 (24.4%)

0.009

CF related liver-disease (n,%)

6 (6.9%)

11 (14.1%)

0.12

P.aeruginosa colonization (n,%)

54/85 (63.5%)

46/76 (60.5%)

0.69

Anaemia (n,%)

2/80 (2.5%)

7/67 (10.4%)

0.08

Ferritin (µg.L-1)

87.4 ± 69.2

49.7 ± 64.8

< 0.001

TSAT (%)

22.8 ± 8.8

17.5 ± 6.1

< 0.001

CRP (mg.L-1)

5.8 ± 8.2

8 ± 14.3

0.78

FEV1 (n,%)
 > 79%
 50-79%
 30-49%
 <30%

n = 86

50 (58.1%)

17 (19.8%)

19 (22.1%)

0

n = 76

32 (42.1%)

27 (35.5%)

16 (21.1%)

1 (1.3%)

0.05

We thank the reviewer: we made a mistake in the table 1. We made the correction in the manuscript (79 replaced 69%).

  1. Please check Table 1 for typos, e.g. p.PhfeF508del heterozygote and ferritine.

 We made the corrections for typos.

Other than that, the article may be of high interest for the readership of ‘Nutrients’.

We thank the reviewer for the kind comments and hope our revisions will be found suitable for publication.

Reviewer 2 Report

Thank you for the opportunity for reviewing the paper. Please, find my comments below.

Major comments:

  1. I am skeptical to including patients with an ongoing acute exacerbation as there is significant variability in ferritin levels in such scenarios. Please, repeat the analyses in a scenario when you excluded all patients meeting the criteria of an acute exacerbation.
  2. Please, perform a systematic search in at least PubMed for studies reporting on the prevalence of iron deficiency anemia in CF and prepare a summary table from the previous studies to contrast the findings of this study.
  3. The flowchart on recruitment should be more detailed (e.g., numbers for cases fulfilling each exclusion criterion should be reported as well; the relevant reporting guideline of the EQUATOR network should be used and referenced in the manuscript). No patients were excluded due to iron replacement therapy? I am not really sure, did 170 out of 415 patients present the clinic within the reported interval of recruitment? Please, clarify the corresponding figure.
  4. Was there a study protocol uploaded to any repository?
  5. Conclusions should not be far-reaching as the ORs scatter around 2. Also, current recommendations on ID screening should be contrasted to current results as well.

Minor comments:

  1. Please, avoid using the words 'modern', 'old', and similar ones as they sound degrading to previous works. Simply, we have past and recent data.
  2. In the discussion section, the word 'large-scale' sounds like an exaggeration.
  3. I suspect that multivariate analysis lacks statistical power with such numbers. Could you please, comment on this issue?
  4. What does prospective mean in this context? Please, clarify ('prospectively collected data' compared to what?). If this is a cross-sectional study, please, identify it so. The word 'single-center' should be preferred over 'monocentric'.
  5. Please, collect study limitations in a paragraph at the end of the discussion section.

Author Response

Thank you for the opportunity for reviewing the paper. Please, find my comments below.

Major comments:

  1. I am skeptical to including patients with an ongoing acute exacerbation as there is significant variability in ferritin levels in such scenarios. Please, repeat the analyses in a scenario when you excluded all patients meeting the criteria of an acute exacerbation.

We agree with the reviewer that exacerbation influence serum ferritin level. Indeed, the systemic inflammation through the increase of inflammatory biomarkers (noticeably interleukin-6) induce the elevation of ferritin level. However, inflammatory syndrome can be present even in the absence of pulmonary exacerbation [1]

No consensus about pulmonary exacerbation of CF exist, but most clinicians use the following symptoms to identify one (Fuchs criteria [2]):

  • New or increased cough, sputum production, chest congestion.
  • Decreased exercise tolerance, increased dyspnoea
  • Increased fatigue, decreased appetite
  • Increased respiratory rate or dyspnoea at rest
  • Change in sputum appearance
  • Fever

In our study, clinically apparent exacerbation was an exclusion criterion. To make this clearer, we added it in the section 2.1 Patients:

Clinically apparent exacerbation of CF between the previous 7 days and the day of the medical consultation defined by one or more of the following symptoms (Fuchs criteria [13]:

-           New or increased cough, sputum production or chest congestion.

-           Decreased exercise tolerance, increased dyspnoea.

-           Increased fatigue, decreased appetite.

-           Increase respiratory rate or dyspnoea at rest.

-           Change in sputum appearance.

-           Fever.

Nevertheless, we recognize that some patients may have experienced an exacerbation more than seven days before the visit that could be responsible for a mild elevation of CRP the day of the annual evaluation. In total, 32 patients (15 women, 17 men) had CRP > 10 mg.L-1 (mean 21.7 ± 17.5). However, among them, 12 women and 4 men were iron deficient according to their ferritin or TSAT level irrespective of the CRP level. Only 9 patients (mean CRP 22.1 ± 19.4) were diagnosed as ID according with the increase of ferritin threshold to 100 µg/L (mean ferritin 66.4 ± 24.3 µg/L).

After excluding these 9 patients, we repeated the analysis with the 64 patients with “absolute ID” (defined by ferritin < 20 µg/L for women and 30 µg/L for men or TSAT < 16% irrespective of gender) and found similar results:

  • A significant association of ID with younger age (29.3 ± 8.9 vs 2 ± 8.8, p = 0.04), age < 30 years (p = 0.03), and female sex (p = 0.002). A trend with P.aeruginosa colonization was found (p = 0.05)
  • The BMI was lower in patients with absolute ID (20.6 ± 2.3 vs2 ± 2.5) but the difference did not reach statistical significance, probably due to the decreased power with a lower sample.

Also, in multivariate analysis we identified the same risk factors, however with larger CI due to the decreased statistical power with a decreased sample size:

  • aeruginosa (OR 2.15, CI 1.03-4.48, p = 0.04), female sex (OR 3.38, CI 1.65-6.93, p = 0.001) and age below 30 years (OR 2.57, CI 1.26-5.27, p = 0.009) were associated with ID.
  • Diabetes mellitus (OR 1.84, CI 0.72-4.69, p = 0.20) and BMI < 20 kg/m² (OR 0.88, CI 0.43-1.78, p = 0.71) were not associated to ID.

As such, to improve the readability and understand ability of our manuscript, we added these information’s in the following paragraphs:

Section 3.2 Iron deficiency prevalence:

“Among them, only 9/73 were classified as ID because the ferritin level was ≤ 100 µg/L with CRP ≥ 10 mg/L”

Section 3.3.2 Multivariate analysis.

“Noticeably, we repeated the multivariate analysis including only the 64 patients defined as ID with low ferritin or TSAT level and excluding those with CRP ≤ 100 µg/L in the case of systemic inflammation (CRP ≥ 10 mg/L) or TSAT ≤ 16%. Similar results were found: ID was significantly associated with P.aeruginosa colonization (OR 2.15, CI 1.03-4.48, p = 0.04), female sex (OR 3.38, CI 1.65-6.93, p = 0.001) and age < 30 years (OR 2.57, CI 1.26-5.27, p = 0.009) whereas diabetes mellitus (OR 1.84, CI 0.72-4.69, p = 0.20) and BMI < 20 kg/m² (OR 0.88, CI 0.43-1.78, p = 0.71) were not associated to ID.”

  1. Please, perform a systematic search in at least PubMed for studies reporting on the prevalence of iron deficiency anemia in CF and prepare a summary table from the previous studies to contrast the findings of this study.

We thank the reviewer for this relevant suggestion. We performed a Pubmed search using the Mesh terms “iron” OR “iron deficiency” AND “cystic fibrosis”, finding 127 articles. After abstract checking, we retained 14 studies, characterized in table 3 as follows:

n

Patients

ID biological
definition

ID
prevalence

Exacerbation

Gettle,
2020 [3]

67

A

ferritin < 12 µg/L
and/or TSAT < 16%

41.8%

PEx+

Kałużna-Czyż, 2018 [4]

46

P

ferritin < 12 µg/L (< 5 yo)

ferritin < 15 µg/L (>5 yo)

39%

PEx+
and PEx-

Yadav, 2014 [5]

27

P

SI < 4 µmol/L

48.1%

PEx+
and PEx-

Gifford, 2012 [6]

12

A

SI < 12 µmol/L

83%

PEx+

Gifford, 2011 [7]

39

A

SI < 12 µmol/L

76.9%

PEx+
and PEx-

von Drygalski, 2008 [8]

26

A+P

SI ≤ 40 µg/dL or

TSAT ≤ 20% or

ferritin ≤ 35 µg/L

61%*

87.5%#

NA

Khalid, 2007 [9]

127

A

ferritin < 12 µg/L (women) and 20 µg/L (men) or

SI < 12 µmol/L or

TSAT ≤ 15% or

sTfR < 1.74 mg/L

18.9% (ferritin)

42.5% (TSAT)

15% (sTfR)

PEx+
and PEx-

Reid, 2002 [10]

30

A

SI < 12 µmol/L
or TSAT < 16%

74%

PEx-

Jaffe, 2002 [11]

144

P

NA

58%

NA

Keevil, 2000 [12]

70

A

ferritin < 12 µg/L (women)

ferritin < 20 µg/L (men)

SI < 12 µmol/L

TSAT < 16%

sTfR < 1.74 mg/L

11% (ferritin)

69% (TSAT)

29% (sTfR)

NA

Pond, 1996 [13]

71

A

TSAT < 16%

62%

NA

Zempsky, 1989 [14]

13

A

ferritin ≤ 25 µg/L

38.4%

PEx-

Ehrhardt, 1987 [15]

127

A+P

ferritin < 12 µg/L

32.3%

PEx+
and PEx-

Ater, 1983 [16]

39

A+P

ferritin < 12 µg/L

SI < 40 µg/dL

TSAT < 16%

33% (ferritin)

25% (SI)

28% (TSAT)

PEx+
and Pex-

* among anaemic patients; # among non-anaemic patients.

A: adult population; n: sample size; NA: not available; P: paediatric population; Pex+: included patients with clinically apparent pulmonary exacerbation; Pex -: included patients without apparent pulmonary exacerbation; SI: serum iron; sTfR: soluble transferrin receptor; TSAT: transferrin saturation.

  1. Cantin, A.M.; Hartl, D.; Konstan, M.W.; Chmiel, J.F. Inflammation in Cystic Fibrosis Lung Disease: Pathogenesis and Therapy. Journal of Cystic Fibrosis 2015, 14, 419–430, doi:10.1016/j.jcf.2015.03.003.
  2. Fuchs, H.J.; Borowitz, D.S.; Christiansen, D.H.; Morris, E.M.; Nash, M.L.; Ramsey, B.W.; Rosenstein, B.J.; Smith, A.L.; Wohl, M.E. Effect of Aerosolized Recombinant Human DNase on Exacerbations of Respiratory Symptoms and on Pulmonary Function in Patients with Cystic Fibrosis. The Pulmozyme Study Group. N Engl J Med 1994, 331, 637–642, doi:10.1056/NEJM199409083311003.
  3. Gettle, L.S.; Harden, A.; Bridges, M.; Albon, D. Prevalence and Risk Factors for Iron Deficiency in Adults With Cystic Fibrosis. Nutrition in Clinical Practice 2020, 35, 1101–1109, doi:https://doi.org/10.1002/ncp.10454.
  4. Kałużna-Czyż, M.; Grzybowska-Chlebowczyk, U.; Woś, H.; Więcek, S. Serum Hepcidin Level as a Marker of Iron Status in Children with Cystic Fibrosis. Mediators Inflamm. 2018, 2018, 3040346, doi:10.1155/2018/3040346.
  5. Yadav, K.; Singh, M.; Angurana, S.K.; Attri, S.V.; Sharma, G.; Tageja, M.; Bhalla, A.K. Evaluation of Micronutrient Profile of North Indian Children with Cystic Fibrosis: A Case–Control Study. Pediatr Res 2014, 75, 762–766, doi:10.1038/pr.2014.30.
  6. Gifford, A.H.; Moulton, L.A.; Dorman, D.B.; Olbina, G.; Westerman, M.; Parker, H.W.; Stanton, B.A.; O’Toole, G.A. Iron Homeostasis during Cystic Fibrosis Pulmonary Exacerbation. Clinical and Translational Science 2012, 5, 368–373, doi:https://doi.org/10.1111/j.1752-8062.2012.00417.x.
  7. Gifford, A.H.; Miller, S.D.; Jackson, B.P.; Hampton, T.H.; O’Toole, G.A.; Stanton, B.A.; Parker, H.W. Iron and CF-Related Anemia: Expanding Clinical and Biochemical Relationships. Pediatr Pulmonol 2011, 46, 160–165, doi:10.1002/ppul.21335.
  8. von Drygalski, A.; Biller, J. Anemia in Cystic Fibrosis: Incidence, Mechanisms, and Association with Pulmonary Function and Vitamin Deficiency. Nutr Clin Pract 2008, 23, 557–563, doi:10.1177/0884533608323426.
  9. Khalid, S.; McGrowder, D.; Kemp, M.; Johnson, P. The Use of Soluble Transferin Receptor to Assess Iron Deficiency in Adults with Cystic Fibrosis. Clinica Chimica Acta 2007, 378, 194–200, doi:10.1016/j.cca.2006.11.021.
  10. Reid, D.W.; Withers, N.J.; Francis, L.; Wilson, J.W.; Kotsimbos, T.C. Iron Deficiency in Cystic Fibrosis: Relationship to Lung Disease Severity and Chronic Pseudomonas Aeruginosa Infection. Chest 2002, 121, 48–54, doi:10.1378/chest.121.1.48.
  11. Jaffé, A.; Buchdahl, R.; Bush, A.; Balfour-Lynn, I.M. Are Annual Blood Tests in Preschool Cystic Fibrosis Patients Worthwhile? Arch Dis Child 2002, 87, 518–520, doi:10.1136/adc.87.6.518.
  12. Keevil, B.; Rowlands, D.; Burton, I.; Webb, A.K. Assessment of Iron Status in Cystic Fibrosis Patients. Ann Clin Biochem 2000, 37 ( Pt 5), 662–665, doi:10.1258/0004563001899708.
  13. Pond, M.N.; Morton, A.M.; Conway, S.P. Functional Iron Deficiency in Adults with Cystic Fibrosis. Respir Med 1996, 90, 409–413, doi:10.1016/s0954-6111(96)90114-6.
  14. Zempsky, W.T.; Rosenstein, B.J.; Carroll, J.A.; Oski, F.A. Effect of Pancreatic Enzyme Supplements on Iron Absorption. American Journal of Diseases of Children 1989, 143, 969–972, doi:10.1001/archpedi.1989.02150200131032.
  15. Ehrhardt, P.; Miller, M.G.; Littlewood, J.M. Iron Deficiency in Cystic Fibrosis. Arch Dis Child 1987, 62, 185–187.
  16. Ater, J.L.; Herbst, J.J.; Landaw, S.A.; O’Brien, R.T. Relative Anemia and Iron Deficiency in Cystic Fibrosis. Pediatrics 1983, 71, 810–814.

  1. The flowchart on recruitment should be more detailed (e.g., numbers for cases fulfilling each exclusion criterion should be reported as well; the relevant reporting guideline of the EQUATOR network should be used and referenced in the manuscript). No patients were excluded due to iron replacement therapy? I am not really sure, did 170 out of 415 patients present the clinic within the reported interval of recruitment? Please, clarify the corresponding figure.

We thank the reviewer to five us the opportunity to improve the flowchart of our study. Our total cohort is 415 patients followed in our adult reference centre. During the 21 weeks of our study, 226 patients appointed to our centre for annual evaluation and were assessed for eligibility. 56 met the exclusion criteria (clinically apparent pulmonary exacerbation in the 7 previous days (n=49) and current or past iron supplementation n=7). Finally, 170 were included but 5 patients had to be excluded because iron biological markers were lacking or uninterpretable due to haemolysis.

We provided a new flow chart and referenced the STROBE guideline.

  1. Was there a study protocol uploaded to any repository?

The study protocol was registered on clinicaltrials : https://clinicaltrials.gov/ct2/show/NCT04584489

  1. Conclusions should not be far-reaching as the ORs scatter around 2. Also, current recommendations on ID screening should be contrasted to current results as well.

We thank the reviewer for this thoughtful suggestion. We mitigated our conclusion and balanced it with the current recommendations on ID screening in pwCF. Here is the new conclusion paragraph:

ID is highly prevalent in adult patients with CF. The current guidelines for pwCF recommends to review iron status annually [40] but a recent survey among CF clinicians showed that few centres are routinely screening for ID. To improve ID screening in pwCF, our results suggest that a special attention should be directed to young women with low BMI and P.aeruginosa colonization. However, male patients would also benefit from ID screening as a much higher prevalence was identified compared to young men in the general population. The therapeutic revolution of CFTR modulators could dramatically improve the prevalence of ID in CF but further prospective studies are required to deter-mine the effect of inflammatory syndrome normalisation on iron metabolism.

Minor comments:

  1. Please, avoid using the words 'modern', 'old', and similar ones as they sound degrading to previous works. Simply, we have past and recent data.

We proceeded to the correction.

  1. In the discussion section, the word 'large-scale' sounds like an exaggeration.

We deleted this qualifying term.

  1. I suspect that multivariate analysis lacks statistical power with such numbers. Could you please, comment on this issue?

We thank the reviewer for the helpful and interesting comment. We agree larger samples are better than smaller samples because they tend to minimize the probability of errors, to maximize the accuracy of population estimates, and to increase the generalizability of the results. Numerous rules-of-thumb have been suggested for determining the minimum number of subjects required to conduct multiple regression analyses but they are heterogeneous and often with minimal empirical evidence. This is problematic because statistical procedures that create optimized combinations of variables (such as multiple regression) tend to overfit the data. Thus, this overfitting can result in erroneous conclusions if models fit to one data set are applied to others. 

For multiple (linear or logistic) regression texts, some authors (e.g., Pedhazur, Multiple Regression in Behavioral Research: Explanation and Prediction. Fort Worth, TX: Harcourt Brace College Publishers 1997, Harris, A primer of multivariate statistics (2nd ed.). New York: Academic Press 1985, Hair et al., Multivariate Data Analysis, 5th Edition, Prentice Hall, New Jersey; 1998, Green, How many subjects does it take to do a regression analysis? Multivariate Behavioral Research 1991; 26: 499-510) suggest subject to variable ratios of 15:1 or 30:1 when generalization is critical. Considering the results reported in our work, multivariate analysis met these rules of thumbs, with 5 predictive factors.

We agree our results need confirmation in larger study but can represent a first step with a moderate statistical power.

However, according to works proposed by Tosteson et al. (Power and sample size calculations for generalized regression models with covariate measurement error. Stat Med. 2003) and Demidenko (Sample size determination for logistic regression revisited. Stat Med. 2007 and Sample size and optimal design for logistic regression with binary interaction. Stat Med. 2008), our statistical power was greater than 80%. Furthermore, according to the reviewer’s comment, we have applied a bootstrap approach on our multivariate analysis in order to guaranty that conclusions are appropriately supported by the results.

This point has been added in the Statistics section (see below):

Categorical data were described with numbers (percentages). Continuous data were expressed as mean and standard-deviation or median and interquartile range, according to the statistical distribution. The assumption normality of the data was assessed using the Shapiro-Wilk test. The comparisons between independent groups according to ID were performed using the Student’s t-test or the Mann-Whitney test when the assumptions of the t-test were not met. Homoscedasticity was assessed using the Fisher-Snedecor test. Categorical data were compared between groups (ID yes/no) using chi-squared or Fisher’s exact tests. Then, multivariable analysis was conducted to determine factors associated with ID. Generalized linear modelling (logistic for binary dependent outcome: ID yes/no) was performed with covariates fixed according to the univariate results and to the clinical relevance, with a particular attention paid on multicollinearity. Furthermore, age and BMI, which did not follow a Gaussian distribution, were categorized according to their statistical distribution. The results were expressed using odds-ratios (OR) and 95% confidence intervals. To ensure the robustness of our results, the final model was validated by a two-step bootstrapping process. In each step, 1,000 bootstrap samples with replacements were created from the training set. In the first one, using the stepwise procedure, we determined the percentage of models including each of the initial variables. In the second step, we independently estimated the logistic model parameters of the final model. The bootstrap estimates of each covariate coefficient and standard errors were averaged from those replicates. Statistical analyses were performed using Stata 15 (StataCorp, College Station, US). All statistical tests were two-sided, with a type I error set at 5%.

  1. What does prospective mean in this context? Please, clarify ('prospectively collected data' compared to what?). If this is a cross-sectional study, please, identify it so. The word 'single-center' should be preferred over 'monocentric'.

Prospectively collected data was to clearly identify that biological markers were collected during the annual visit for systematic medical check-up, in contrast to other studies where the iron status was extracted from the medical record.

We added the words cross-sectional and single-centre over monocentric: as such, the title of the manuscript was modify in: Iron deficiency in cystic fibrosis: a prospective cross-sectional single-centre study in a referral adult centre.

  1. Please, collect study limitations in a paragraph at the end of the discussion section.

We collected the main limitations of our work in a final paragraph at the end of the discussion section.

Some limitations of our study can be raised. First our study is a single centre and took place in a referral centre: thus, our results might not be generalizable to the whole pwCF population. Furthermore, our results are only applicable to adult patients and specific studies should be conducted in the paediatric population. Second, although we included 40% of our cohort, the sample size may limit multiple comparison and subgroup analyses. Third, the cross-sectional nature of our study does not allow us to study the effect of treatments, such as antibiotics or CFTR modulators on the incidence of ID. To the best of our knowledge, our study is the largest published to date, representing a first step analyse with moderate statistical power, providing basis and hypothesis for future prospective studies to confirm these results in larger samples.

Round 2

Reviewer 2 Report

The authors commented on and reacted to my points adequately. I have no further questions.